# Study on the Relevance of Metabolic Syndrome and Incidence of Gastric Cancer in Korea

**DOI:** 10.3390/ijerph16071101

**Published:** 2019-03-27

**Authors:** HaiWon Yoo, Hyeongsu Kim, Jung-Hyun Lee, Kun-Sei Lee, Min-Jung Choi, Hye-Ry Song, Jung-Hee Cho, Jin-Hyeong Kim

**Affiliations:** Department of Preventive Medicine, School of Medicine, Konkuk University, Seoul 05029, Korea; ilkimsh@hanmail.net (H.Y.); leedouble@nate.com (J.-H.L.); leekonkuk@gmail.com (K.-S.L.); cmj17kr@gmail.com (M.-J.C.); sss3693@naver.com (H.-R.S.); jhee2725@gmail.com (J.-H.C.); wlsgud1026@naver.com (J.-H.K.)

**Keywords:** metabolic syndrome, gastric cancer, relevance, big data

## Abstract

(1) Background: This study aimed to determine the relevance between stages of metabolic syndrome (MS) progression and the incidence of gastric cancer utilizing a big data cohort for the national health checkup. (2) Methods: There were 7,785,098 study subjects, and three stages of metabolic syndrome were categorized using the health checkup results from 2009. Incidence of gastric cancer was traced and observed from the date of the health insurance benefit claim in 2009 until 31 December, 2016, and Cox hazard-proportional regression was performed to determine the risk of gastric cancer incidence based on the stage of progression for metabolic syndrome. (3) Results: Hazard ratio (HR) incidence rate for the MS group was 2.31 times higher than the normal group (95% CI 2.22–2.40) after adjustment (Model 4). The HR incidence rate of gastric cancer for the pre-MS group was 1.08 times higher (95% CI 1.04–1.12) than the normal group, while the HR incidence rate of gastric cancer for the MS group was 1.26 times higher (95% CI 1.2–1.32). (4) Conclusions: Causal relevance observed in this study between metabolic syndrome and incidence of gastric cancer was high. Promotion and education for active responses in the general population and establishment of appropriate metabolic syndrome management systems to prevent gastric cancer are needed.

## 1. Introduction

Metabolic syndrome (MS) is a disease group stemming from poor lifestyle habits. Metabolic syndrome, which is defined by the presence of at least three out of the five factors including abdominal obesity, elevated triglycerides, low high-density lipoprotein cholesterol (HDL), high blood pressure, and high fasting glucose [1], is becoming an almost ubiquitous severe health issue across the globe [2]. Prevalence rates of metabolic syndrome worldwide varies depending on race, environmental factors, age and gender composition of population, genetic differences, physical activity level, eating habits, and difference in measurement standards [3]. Prevalence rates of metabolic syndrome in Korea has increased continuously from 24.9% in 1998 to 29.2% in 2001, 30.4% in 2005, and 31.3% in 2007 [4]. 

Metabolic syndrome is known to increase the risk of type 2 diabetes, cardiovascular diseases, and all causes of death [5]. Studies on metabolic syndrome related to incidence of liver cancer, colorectal cancer, pancreatic cancer, and breast cancer post-menopause and its effect on subsequent death are being performed [6,7,8]. Also, metabolic syndrome has been reported to increase the risk of gastric adenocarcinoma [9], and it has been reported that waist circumference (abdominal obesity) specifically, among metabolic syndrome factors, is related to increase in risk of gastric adenocarcinoma [10]. Previous studies investigated gastric cancers, but in association with diabetes, and results were conflicting [11]; however, a meta-analysis of diabetic patients also found that the risk of gastric cancer increased slightly [11]. On the other hand, there is no study on the risk of hypertension, serum levels of triglycerides or cholesterol, and gastric cancer related to metabolic syndrome [12].

According to the Global Cancer Report (GLOBOCAN) of the International Agency for Research on Cancer (IARC), about 39.6 per 100,000 people in Korea have gastric cancer, and the occurrence of gastric cancer ranks first in the world [13]. Although there have been many studies on the risk factors for gastric cancer, there have been no studies related to metabolic syndrome in Korea. Recently, studies on metabolic syndrome and cancer incidence have been conducted in Korea [14,15], but these studies were conducted on people who received health checkups at university hospitals, so it was difficult to expand the findings to the general population. They also had limitations of short observation periods to confirm significance of metabolic syndrome in the incidence of cancer and a lack in the number of cancer cases to calculate significant results.

This study aimed to determine the relevance between metabolic syndrome and incidence of gastric cancer through utilization of a national health big data cohort, which can represent the general population group in Korea.

## 2. Materials and Methods

### 2.1. Study Design and Sample

This study is a retrospective follow-up study analyzing the data on the national health checkup and insurance benefit claim from the National Health Insurance Corporation to determine the relevance between metabolic syndrome and incidence of gastric cancer. National health checkup data from 2009 was used for stages of progression of metabolic syndrome among cases, and insurance benefit claim data was used for determining the incidence of gastric cancer. The observation period was from the date of health checkup in 2009 until 31 December 2016 (Figure 1). A total of 15,036,607 people were subject to a general health checkup in 2009, but only 9,927,538 people participated in the health checkup. The following people were excluded from the study:(1)Cases below the age of 30.(2)Cases with missing values among metabolic syndrome checkup items (fasting blood sugar, systolic and diastolic blood pressure, TG, HDL, waist circumference).(3)Cases with a history of cardiovascular or cerebrovascular diseases and cancer; beneficiaries for cancer (C00-97), cardiovascular/cerebrovascular diseases (ischemic heart disease I20-25, cardiac failure I50 or I42), cerebrovascular disease (I60-69), atrial fibrillation (I48), and circulatory system disease (I00-99).

Therefore, a total of 7,785,098 study subjects were divided into a normal group, a pre-MS group, and an MS group, which consisted of 2,592,741 people; 3,948,504 people; and 1,243,853 people, respectively (Figure 1).

### 2.2. Measurements

#### 2.2.1. Dependent Variable: Gastric Cancer

Topography was coded according to the International Classification of Diseases-10th Revision (ICD-10). Those who were diagnosed with ICD-10 code C16 during the course of disease development were classified as gastric cancer patients. Gastric cancer events were defined as an incident case after 2009; all cases with incident cancers before 2009 were excluded from the analysis.

#### 2.2.2. Independent Variable: Metabolic Syndrome

Criteria for diagnosis of metabolic syndrome within the general health checkup for the National Health Insurance Corporation of Korea adhered to National Cholesterol Education Programme Adult Treatment Panel III (NCEP-ATPIII), which reflected the waist measurement standard for Koreans, and this study also defined the criteria for diagnosis of metabolic syndrome with the same criteria. Criteria for diagnosis of metabolic syndrome by cause is shown in Table 1. Subjects below criteria values for all five metabolic syndrome factors in the 2009 health checkup were categorized into the normal group, while subjects with one to two factors above criteria values were categorized into the pre-MS group, and subjects with three or more factors above criteria values were categorized into the MS group.

#### 2.2.3. Adjusted Variables: Sex, Age, Health Behavior, Family History, Laboratory Findings

Gender, age, health behavior, family history, and laboratory findings, which were confirmed at the point of checkup, were utilized for adjusted variables. Age was divided into ten years. Smoking, drinking, and physical activities within the health survey were selected for health behavior. Smoking was categorized into three groups that consisted of not smoking, smoked in the past but no longer, and currently smoking. Alcoholic drinking was categorized based on frequency into four groups consisting of does not drink, two to three times per month, one to four times per week, and almost every day. Physical activity was categorized based on frequency into three groups consisting of none, one to four times per week, and almost every day. Family history was categorized into high blood pressure, diabetes, cerebral stroke, ischemic heart disease, and cancer (Y/N). Hemoglobin, total cholesterol, HDL, ALT (Alanine aminotransferase), and creatinine, which are national health checkup items, were used for laboratory finding.

### 2.3. Statistical Analysis

Incidence of gastric cancer and the comparison between frequency and average values of independent variables and adjusted variables were conducted with a chi-square test and t-test, respectively. Cox proportional-hazard regression was applied for the analysis of risk of gastric cancer incidence from metabolic syndrome. Five Cox proportional-hazard models were designed and analyzed, including the unadjusted model depending on the calibration of variables, which directly and indirectly affected the disease and mortality. Gender and age were adjusted for the Cox proportional-hazard Model 1. Health behaviors (smoking, exercise) were adjusted for the Cox proportional-hazard Model 2, while cerebral stroke, high blood pressure, cardiac disease, and diabetes were adjusted for the cox proportional-hazard Model 3. BMI, creatinine, hemoglobin, and other relevant values were added for Cox proportional-hazard Model 4. Results were suggested with a hazardous ratio (HR) and a 95% confidence interval (95% CI). Significance level for all statistical values was set at *p* < 0.05. All data analyses were performed through SAS software (version 9.1; SAS Institute Inc., Cary, NC, USA).

### 2.4. Ethical Consideration

Ethical approval was obtained from the institutional review board of Konkuk University (approval number 7001355-201711-E-054).

## 3. Results

### 3.1. Cumulative Incidence Rate of Gastric Cancer

Cumulative incidence rate of gastric cancer among study subjects was 0.5% (33,929 gastric cancer patients), and cumulative incidence rates of gastric cancer by stage of metabolic syndrome were 0.3%, 0.5%, and 0.7%, respectively, for the normal group, the pre-MS group, and the MS group (*p* < 0.001).

Relevance between other variables and incidence of gastric cancer is shown below in Table 2.

### 3.2. Risk of Gastric Cancer Following the Progression of Metabolic Syndrome

The HR incidence rates of gastric cancer according to the stages of progression for metabolic syndrome (normal, pre-MS, and MS) are shown below in Table 3. Model 4, with the smallest Akaike information criterion (AIC) value of 656,447.46, was selected as the optimal model. The HR incidence rate of gastric cancer for the pre-MS group compared to the normal group before adjustment (unadjusted model) was 1.61 times higher (95% CI 1.56–1.66), while the HR incidence rate for the MS group was 2.31 times higher (95% CI 2.22–2.40). After adjustment (Model 4), the HR incidence rate of gastric cancer for the pre-MS group was 1.08 times higher (95% CI 1.04–1.12) compared to the normal group, while the HR incidence rate of gastric cancer for the MS group was 1.26 times higher (95% CI 1.2–1.32).

## 4. Discussion

This study aimed to suggest objective evidence to support the necessity of actively preventing and managing metabolic syndrome by determining the relevance between the stages of progression for metabolic syndrome and incidence of gastric cancer.

The cumulative incidence rates of gastric cancer by stage of metabolic syndrome (per 10,000 people) were 28 people for the normal group, 46 people for the pre-MS group, and 67 people for the MS group (*p* < 0.001), and it was observed that the incidence rate of gastric cancer increased as metabolic syndrome progressed. After adjustment for some variables, HR incidence rates of gastric cancer were 1.08 times higher (95% CI 1.04–1.12) for the pre-MS group and 1.26 times higher (95% CI 1.2–1.32) for MS group compared to the normal group as metabolic syndrome progressed. This was similar to study results that found metabolic syndrome increased the risk of gastric adenocarcinoma incidence by 1.44 times (1.14–1.82) over 12 years on average based on 192,903 subjects [2], and study results that found metabolic syndrome increased the risk of gastric adenocarcinoma incidence by 1.18 times (1.00–1.38) [15].

In the study by Lin et al. (2016) [2], waist circumference, high blood pressure, and blood glucose level were discussed as major factors increasing the risk of gastric adenocarcinoma, and the risk of gastric cancer incidence was especially high for waist circumference, at 1.71 times higher (1.05–2.80), and high blood pressure, at 2.41 times higher (1.44–4.03). Also, according to the study by Lindkvist et al. (2013) [15], the risk of gastric adenocarcinoma incidence was 5.65 times higher (1.57–20.34) for women with a higher blood glucose level. Another meta-analysis study suggested that diabetes was associated with a significantly increased HR for gastric cancer (HR 1.11, 95% CI 1.00–1.24) [11]. According to Japanese studies in Asian countries, fasting plasma glucose significantly increased the HR for gastric cancer (HR 3.0, 95% CI 1.5–6.4) [16]. The study also described the association between *Helicobacter pylori* and fasting plasma glucose, which affected the occurrence of gastric cancer. The high fasting plasma glucose group infected with *Helicobacter pylori* was at a higher risk of gastric cancer than the low fasting plasma glucose group (HR 4.2, 95% CI 1.6–11.1) [16]. Also, metabolic syndrome has been reported to be related to deaths caused by gastric cancer as well as incidence of gastric cancer. According to a recently conducted study, the median survival time of metabolic syndrome patients is 31.3 months. This was significantly shorter in terms of statistics compared to 157.1 months for the control group (*p* < 0.01), and the mortality rate for patients with metabolic syndrome prior to operation was 2.3 times (2.02–2.62) higher [17]. Several previous studies have shown that metabolic syndrome criteria not only have significantly affected gastric cancer incidence, but they have also increased mortality. In Korea, however, studies that investigate the relationship between each metabolic syndrome criteria and the risk of gastric cancer have not been conducted yet. This study was conducted to show the relevance of metabolic syndrome and gastric cancer as the first step.

Most of the studies on the relevance between metabolic syndrome and gastric cancer were cohort studies taking place in Europe. Meta-analyses on metabolic syndrome and risk of gastric cancer have included Asian nations, but they were studies including only cases from China and Japan [18,19,20,21], and there were almost no studies including the Korean population group. This study secured a sufficient number of gastric cancer patients through follow-up observation over a long period of time. This was utilized to calculate the incidence of gastric cancer and hazard ratios based on the progression of metabolic syndrome. The purpose of such calculation lies in establishing basic data to explain the relationship between metabolic syndrome and incidence of gastric cancer in the future. Also, this study discovered significant relevance between metabolic syndrome and the risk of gastric cancer incidence after adjustment of variables, including family history and blood test results along with social demographic factors, while most studies have not considered many confounding variables.

However, the limitations in this study are the following: First, data for *Helicobacter pylori* infections, gastric ulcers, eating disorders and other known risk factors of gastric cancer could not be obtained, so such adjustments were not made. This is a limitation because of the usage of secondary data. Second, an inability to perform follow-up observations narrowing the risk factors for metabolic syndrome to a single point remains to be the limitation of this study. Therefore, studies with follow-up observations that incorporate the discovery of risk factors for metabolic syndrome in the future are necessary.

## 5. Conclusions

The HR incidence rate of gastric cancer as metabolic syndrome phases progress has increased significantly, compared to that of the normal group, which resulted in high causal relevance between metabolic syndrome and the incidence of gastric cancer. Various studies on the relevance of various types of cancer, including metabolic syndrome and gastric cancer, are necessary in the future. Also, the causal relevance observed in this study between metabolic syndrome and the incidence of gastric cancer was high. Promotion and education for active responses in the general population, and establishment of an appropriate management system for metabolic syndrome to prevent gastric cancer, and cardiovascular and cerebrovascular disorders, is needed.

## Figures and Tables

**Figure 1 ijerph-16-01101-f001:**
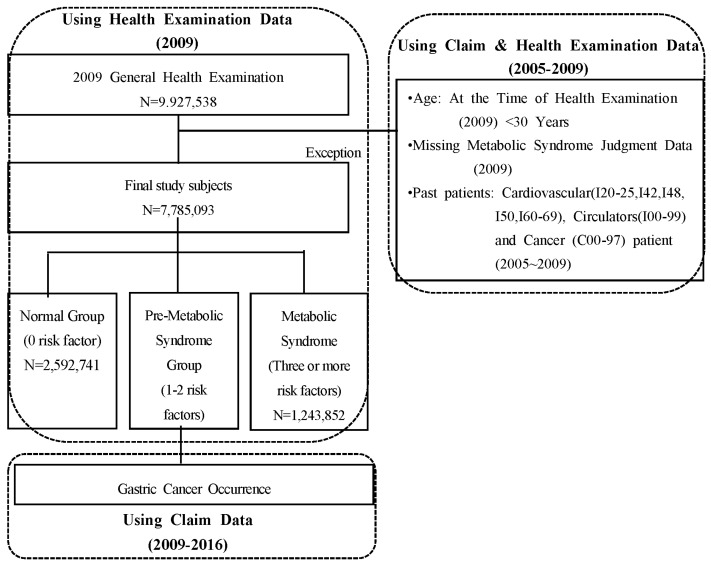
Study design and process of selecting the population.

**Table 1 ijerph-16-01101-t001:** Diagnostic criteria for the clinical diagnosis of metabolic syndrome.

Waist Circumference	Waist Circumference >90 cm in Men and >85 cm in Women
**Blood Pressure**	≥130/85 mm Hg, treatment of previously diagnosed hypertension
**Triglyceride**	≥150 mg/dL (1.69 mmol/L), treatment of previously diagnosed hypertriglyceridemia
**HDL**	<40 mg/dL (1.04 mmol/L) in men and <50 mg/dL (1.29 mmol/L) in women, specific treatment for this lipid abnormality
**Glucose**	≥100 mg/dL (≥5.56 mmol/L), previously diagnosed type 2 diabetes

**Table 2 ijerph-16-01101-t002:** Incidence rate of gastric cancer by variables.

Characteristics	Category	Non-Cancer, N (%)	Gastric Cancer, N (%)	*p*
**Metabolic syndrome (MS) stage**	**Normal**	2,585,352 (99.7)	7389 (2.8)	<0.001
	**Pre-MS**	3,930,296 (99.5)	18,204 (4.6)	
	**MS**	1,235,516 (99.3)	8336 (6.7)	
**Sex**	**Male**	4,405,631 (99.5)	24,010 (5.4)	<0.001
	**Female**	3,345,533 (99.7)	9919 (3.0)	
**Age**	**30–39**	1,511,110 (99.9)	635 (4.0)	<0.001
	**40–49**	2,157,650 (99.8)	3999 (1.8)	
	**50–59**	2,381,187 (99.5)	11,551 (4.8)	
	**≥60**	1,701,217(98.9)	17,744 (10.3)	
**Smoke**	**Non-smoker**	4,367,571(99.7)	14,866 (3.4)	<0.001
	**Ex-smoker**	1,057,292(99.4)	6806 (6.4)	
	**Smoker**	2,278,031(99.5)	12,048 (5.3)	
**Alcohol consumption**	**No drink**	3,455,973 (99.6)	14,092 (4.1)	<0.001
	**2–3/per month**	3,168,394 (99.6)	12,142 (3.8)	
	**1–4/per week**	785,965 (99.4)	4951 (6.3)	
	**≥5/per week**	241,978 (99.0)	2401 (9.8)	
**Physical Exercise, per week**	**No exercise**	3,555,044 (99.6)	15,243 (4.3)	<0.001
	**1–4/per week**	1,568,184 (99.6)	6284 (4.0)	
	**≥5/per week**	2,564,894 (99.5)	12,136 (4.7)	
**FH of HP**	**No**	4,476,882 (99.5)	19,443 (4.3)	<0.001
	**Yes**	823,010 (99.6)	3183 (3.9)	
**FH of DM**	**No**	4,594,193 (99.6)	1972 (4.3)	<0.060
	**Yes**	417,874 (99.5)	2906 (4.1)	
**FH of Stroke**	**No**	4,877,551 (99.6)	20,450 (4.2)	<0.001
	**Yes**	253,439 (99.6)	2182 (5.2)	
**FH of Heart Disease**	**No**	5,038,086 (99.6)	21,552 (4.3)	<0.212
	**Yes**	781,926 (99.4)	1042 (4.1)	
**FH of CA**	**No**	4,516,223 (99.4)	18,326 (4.0)	<0.001
	**Yes**	781,926 (99.4)	4328 (5.5)	
**Total, N (%)**		**7,785,093 (100.0)**	**33,929 (0.5)**	

FH, Family history; HP, Hypertension; DM, Diabetes; CA, Cancer.

**Table 3 ijerph-16-01101-t003:** Hazardous ratio (HR) incidence rates of gastric cancer according to the progression of metabolic syndrome.

Characteristics	Category	HR (95% CI)									
		Non–Adjusted		Model 1		Model 2		Model 3		Model 4	
**MS stage**	**Normal**	**Ref.**		**Ref.**		**Ref.**		**Ref.**		**Ref.**	
	**Pre-MS**	1.61	(1.56 1.66)	1.03	(1.00 1.07)	1.02	(0.99 1.05)	1.03	(0.99 1.06)	1.08	(1.04 1.12)
	**MS**	2.31	(2.22 2.40)	1.14	(1.09 1.18)	1.11	(1.08 1.17)	1.12	(1.08 1.17)	1.26	(1.2 1.32)
**sex**	**female**			Ref.		Ref.		Ref.		Ref.	
	**male**			2.12	(2.06 2.18)	1.79	(1.72 1.86)	1.74	(1.7 1.84)	2.18	(2.08 2.29)
**age**	**30–39**			Ref.	(3.68 4.52)	Ref.		Ref.		Ref.	
	**40–49**			4.07		4.05	(3.66 4.49)	4.03	(3.63 4.46)	4.04	(3.64 4.48)
	**50–59**			11.68	(10.59 12.89)	11.74	(10.64 12.95)	11.46	(10.39 12.56)	11.24	(10.17 12.41)
	**≧60**			25.65	(23.26 28.29)	26.11	(23.68 28.79)	25.31	(22.94 27.93)	24.69	(22.36 27.27)
**Smoke**	**Non-smoker**					Ref.		Ref.		Ref.	
	**Ex-smoker**					1.21	(1.16 1.26)	1.18	(1.13 1.23)	1.18	(1.13 1.23)
	**Smoker**					1.35	(1.3 1.4)	1.30	(1.25 1.35)	1.31	(1.26 1.37)
**Alcohol Consumption**	**No drink**					Ref.		Ref.		Ref.	
	**2–3/per month**					1.00	(0.97 1.03)	1.00	(0.97 1.03)	1.01	(0.97 1.04)
	**1–4/per week**					1.16	(1.11 1.22)	1.17	(0.97 1.03)	1.17	(1.12 1.23)
	**≥5/per week**					1.35	(1.28 1.43)	1.35	(1.28 1.43)	1.34	(1.27 1.42)
**Exercise**	**no exercise**					Ref.		Ref.		Ref.	
	**1–4/per week**					0.98	(0.94 1.01)	0.98	(0.94 1.01)	0.98	(0.94 1.02)
	**≥****5/per week**					0.99	(0.96 1.02)	0.99	(0.96 1.02)	0.99	(0.96 1.02)
**FH of Hypertension**	**no**							Ref.		Ref.	
	**yes**							0.88	(0.84 0.91)	0.88	(0.85 0.92)
**FH of DM**	**no**							Ref.		Ref.	
	**yes**							1.02	(0.98 1.06)	1.03	(0.98 0.99)
**FH of Stroke**	**no**							Ref.		Ref.	
	**yes**							0.95	(0.9 0.99)	0.95	(0.91 1)
**FH of Heart disease**	**no**							Ref.		Ref.	
	**yes**							0.93	(0.87 0.99)	0.93	(0.87 0.99)
**FH of CA**	**no**									Ref.	
	**yes**									1.18	(1.14 1.22
**BMI**										0.99	(0.98 0.99)
**Hemoglobin**										0.89	(0.88 0.9)
**Serum creatinine**										1.01	(1.00 1.01)
**Total cholesterol**										1.00	(1.00 1.00)
**LDL cholesterol**										1.00	(1.00 1.00)
**ALT(SGPT)**										1.00	(1.00 1.00)
**AIC**		675,732.09		659,523.22		659,140.48		658,994.70		656,447.46	

Values are presented as β (95% confidence interval). MS, Metabolic Syndrome; Ref, Reference; FH, Family History; ALT, Alanine aminotransferase; AIC, Akaike Information Criterion; Model 1 adjusted: sex, age; Model 2 adjusted: sex, age smoke, alcohol consumption, and exercise; Model 3 adjusted: sex, age smoke, alcohol consumption, exercise, FH of HP, FH of DM, FH of Stroke, FH of Heart Disease, and FH of CA; Model 4 adjusted: sex, age smoke, alcohol consumption, exercise, FH of HP, FH of DM, FH of Stroke, FH of Heart Disease, FH of CA, Hemoglobin, Serum creatinine, Total cholesterol, LDL cholesterol, and ALT.

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
