# Peer review of "Study on the Relevance of Metabolic Syndrome and Incidence of Gastric Cancer in Korea"

_ijerph, 2019, doi:10.3390/ijerph16071101_

Round 1

Reviewer 1 Report

The research article entitled " Study om relevance of Metabolic Syndrome and Occurrence of Gastric Cancer in Korea" attempts to show direct link between Metabolic Syndrome and Gastric Cancer. But authors should consider few points before their work is finally accepted for publication:

Consider explaining abbreviations, for e.g.  AIC, when these are being used for the first time in text.

Check references, for spacing and formatting. 

Though authors have shown relationship between metabolic syndrome and gastric cancer. They should try to state reason for causal and effect relationship between the two.

Individual contribution of MS factors must be stated while deciding them to be factors responsible for gastric cancer.

Introduction and Discussion needs to be bit extensive to give readers insight of the study.

Statistical significance stated in Table 2 must be self explanatory, stating for which group are p values being provided. 

Author Response

[Hyeong-Su Kim ]

[Department of Preventive Medicine, School of Medicine, Konkuk University]

[120 Neungdong-ro, Gwangjin-gu, Seoul, Korea]

[2019.03.20]

Dear doctor

Thanks for your good comments to improve our manuscript.

I will send you the opinions of the three reviewer and the answers of the authors.

I would like confirmation.

Sincerely,

Hyeong-Su Kim

No

Reviewer   opinion

Author   answer

1

Consider explaining abbreviations, for e.g.  AIC,   when these are being used for the first time in text.

Revised.

(Line 204)

2

Check references, for spacing and formatting

Revised.

3

Though authors have shown relationship between   metabolic syndrome and gastric cancer. They should try to state reason for   causal and effect relationship between the two.

The relationship between metabolic syndrome and gastric cancer   was explained by referring to more previous research results.

(Reference 11-13,16-17 / Line 56~59, 8P Line 19-25)

4

Individual contribution of MS factors must be stated   while deciding them to be factors responsible for gastric cancer.

It is very meaningful research see whether the gastric cancer   and the individual contribution of metabolic syndrome factors.

However, the basic purpose of our study was to see whether   metabolic syndrome was associated with the risk of gastric cancer.

Therefore, we investigated the association of the risk of   developing the metabolic syndrome, not the individual factors.

Also, if the modification period is sufficient, it will be   possible to re-grant permission to access the data, but it will not be   possible now.

This section was added to the content in the body as a   limitation of our study.

5

Introduction and Discussion needs to be bit extensive   to give readers insight of the study.

Revised.

(Line 60-63)

6

Statistical significance stated in Table 2 must be self   explanatory, stating for which group are p values being provided.

Revised.

(Table2)

Reviewer 2 Report

This manuscript has been poorly written with extensive grammar errors. Needs to be rewritten.

Line 43 should say high levels of those factors (e.g., TG, ...) are related to metabolic syndrome. 

Too many small paragraphs are seen in the manuscripts. Some of them that are related should be combined.

Introduction and Discussion need to written stronger by bringing more publications and citations. The association between metabolic syndrome and gastric cancer should be more explained in Introduction. Why this association is important to be studied?

"Age was defined by period of 10 years" what does the sentence mean? Be more clear.

What does OT stand for?

Author Response

[Hyeong-Su Kim ]

[Department of Preventive Medicine, School of Medicine, Konkuk University]

[120 Neungdong-ro, Gwangjin-gu, Seoul, Korea]

[2019.03.20]

Dear doctor

Thanks for your good comments to improve our manuscript.

I will send you the opinions of the three reviewer and the answers of the authors.

I would like confirmation.

Sincerely,

Hyeong-Su Kim

Reviewer2

No

Reviewer opinion

Author answer

1

This manuscript has been poorly written with   extensive grammar errors. Needs to be rewritten

Revised

2

Line 43 should say high levels of those factors   (e.g., TG, ...) are related to metabolic syndrome. 

Revised

3

Too many small paragraphs   are seen in the manuscripts. Some of them that are related should be combined

Revised

4

Introduction and Discussion   need to written stronger by bringing more publications and citations. The   association between metabolic syndrome and gastric cancer should be more   explained in Introduction. Why this association is important to be studied?

Revised

(Line 56~59, 8P Line 19-25)

5

"Age was defined by period of 10 years"   what does the sentence mean? Be more clear.

Revised

(Line157)

6

What does OT stand for?

Revised

Reviewer 3 Report

It is an interesting paper where the authors try to associate the metabolic syndrome with gastric cancer. However, there are some points that must be corrected.

As the authors mention, one of the limitations of the study is the data of H. pylori infection. For this reason the authors cannot associate the metabolic syndrome with gastric cancer (3.2 Risk of gastric cancer following the progression of metabolic syndrome). I think it is better to report the frequency of gastric cancer with each variable. I suggest change the word “occurrence” by “frequency or incident”

The authors must include the meaning of AIC and IC. “Model 4 with the smallest AIC value”

Table 2

The gastric cancer group should not be included in the subtotal group. i.e. n= XX no gastric cancer and n= XX gastric cancer

Table 3

At the foot of the table should be include the variables of model 1, model 2, model 3 and model 4.

Author Response

[Hyeong-Su Kim ]

[Department of Preventive Medicine, School of Medicine, Konkuk University]

[120 Neungdong-ro, Gwangjin-gu, Seoul, Korea]

[2019.03.20]

Dear doctor

Thanks for your good comments to improve our manuscript.

I will send you the opinions of the three reviewer and the answers of the authors.

I would like confirmation.

Sincerely,

Hyeong-Su Kim

Reviewer3

No

Reviewer opinion

Author answer

1

As the authors mention, one of the limitations of the study is the data   of H. pylori infection. For this reason the authors cannot   associate the metabolic syndrome with gastric cancer (3.2   Risk of gastric cancer following the progression of metabolic syndrome).

Referring   to the preceding studies, the association between metabolic syndrome and 'H.   pylori infection' was described in the discussion section.

(8p Line22-25)

2

I think it is better to report the frequency of gastric cancer with each   variable. I suggest change the word “occurrence” by “frequency or incident”

We   have changed the term from 'occurrence' to 'incident' according to your   opinion.

3

The authors must include the meaning of AIC and IC. “Model 4 with the   smallest AIC value”

Revised

4

Table 2. The gastric cancer group should not be included in the subtotal   group. i.e. n= XX no gastric cancer and n= XX gastric cancer

Revised

5

Table   3. At the foot of the table should be include the variables of model 1, model   2, model 3 and model 4.

Revised

Round 2

Reviewer 2 Report

The changes that you've made have improved the manuscript however, there still are many writing issues that can disturb reading of this manuscript and decrease the interest of readers. I recommend that this manuscript needs to be revised by a English professional.